# Clinical Features and Laboratory Findings of Hospitalized Children with Infectious Mononucleosis Caused by Epstein–Barr Virus from Croatia

**DOI:** 10.3390/pathogens14040374

**Published:** 2025-04-10

**Authors:** Laura Prtorić, Ante Šokota, Silvana Karabatić Knezović, Goran Tešović, Snjezana Zidovec-Lepej

**Affiliations:** 1Pediatric Infectious Diseases Department, Clinical Hospital Center Rijeka, 51 000 Rijeka, Croatia; laurapp3@gmail.com; 2Pediatric Infectious Diseases Department, University Hospital for Infectious Diseases, 10 000 Zagreb, Croatia; sokota.ante@gmail.com; 3Secondary Medical School Split, Šoltanska 15, 21 000 Split, Croatia; silvana.karabatic-knezovic@skole.hr; 4School of Medicine, University of Zagreb, Šalata 2, 10 000 Zagreb, Croatia; gtestovic@bfm.com; 5Department of Immunological and Molecular Diagnostics, University Hospital for Infectious Diseases, 10 000 Zagreb, Croatia

**Keywords:** Epstein–Barr virus, infectious mononucleosis, children, complications

## Abstract

The aim of this retrospective 6-year study was to analyze demographic, laboratory and clinical features of 212 patients (<18 years of age) with EBV-associated infectious mononucleosis (IM) hospitalized in a tertiary clinical care center in southeastern Europe and to identify possible predictors of complications. The median patient age was 14.7 years (IQR 7.7–16.5 years), with 59.4% of patients aged between 13 and 18 years. A total of 51.2% of patients were hospitalized within 7 days following the onset of symptoms (median duration of hospitalization was 9 days, IQR 7–11 days). The most common symptoms included fever (97.16%), tonsillitis (87.3%), lymphadenopathy (79.2%), hepatomegaly (77.4%) and splenomegaly (73.1%). Symptom distribution, maximal fever and fever duration did not differ among different age groups. The most common complications included tonsillar hypertrophy, thrombocytopenia, anemia, neutropenia and leukopenia but all patients showed favorable outcomes. Patients who developed three or more complications and those presenting with thrombocytopenia showed significantly longer hospitalization durations. Platelet count, bilirubin, ESR and AST were identified as the most accurate predictors of hospitalization duration using multiple linear regression analysis. Therefore, our results suggest that clinical assessment of individual patients remains the most reliable parameter for patient management and that laboratory findings play only a supporting role.

## 1. Introduction

Epstein–Barr virus (EBV) or human herpesvirus 4 is an oncogenic DNA virus that belongs to the family *Orthoherpesviridae*, subfamily *Gammaherpesvirinae* and genus Lymphocryptovirus [1]. The virus is usually transmitted via body fluids, especially saliva, but transfusion-mediated transmission in severely immunosuppressed patients as well as transmission via transplantation have also been described [2,3,4]. EBV initially infects epithelial and resting B-lymphocytes in the oropharynx and initiates a lytic replication cycle that allows it to spread throughout the host. The key feature of the immune response during primary EBV infection is an extensive activation of CD8+ T-cells and NK-cells that restricts viral replication and can be associated with clinical manifestations of infectious mononucleosis (IM). Subsequently, the virus establishes a lifelong latent infection in B-lymphocytes that is strictly regulated by transcriptional modulation of viral gene expression and can be associated with malignant transformation of cells. Therefore, EBV is considered an important etiological cofactor for the development of a wide range of malignancies including Burkett’s lymphoma, post-transplant lymphoproliferative diseases, nasopharyngeal carcinoma, Hodgkin’s lymphoma, and gastric carcinoma as well as non-malignant but chronic illnesses [5,6]. Despite the fact that EBV infection is associated with considerable morbidity and mortality on the global level, licensed prophylactic or therapeutic vaccines are still not available [7].

Primary EBV infection in childhood is typically subclinical but it can lead to the development of IM in up to 70% of adolescents and young adults aged 15–24 years [8]. Typical signs and symptoms of IM include fever, fatigue, sore throat, head and body aches, swollen lymph nodes in the neck and hepatosplenomegaly [9,10]. Patients with IM occasionally present with a nonpruritic, faint rash that is probably caused directly by the virus, and pruritic rash, often observed following treatment with beta-lactam antibiotics, mainly ampicillin or amoxicillin [11]. The most common laboratory findings in IM include lymphocytosis, atypical lymphocytes and increased values of liver transaminases [10]. The risk of severe symptoms is positively correlated with the age of the patient at the time of primary infection [12]. In routine practice, diagnosis of IM is mainly based on clinical presentation that includes a classical symptomatic triad of pharyngitis and cervical lymphadenopathy with lymphocytosis with supporting etiological diagnostics based on serology and, in cases of early diagnosis or inconclusive serology results, real-time PCR.

Although IM is generally benign and self-limiting, it can lead to serious complications affecting various systems, including respiratory, cardiovascular, hematological, genitourinary, gastrointestinal, neurological, and psychiatric. It can also increase the risk of autoimmunity, allergies, neoplasms, and X-linked lymphoproliferative disease [13]. Rare but severe complications include upper airway obstruction (1–3.5% of patients) and spontaneous splenic rupture (0.1–0.5% of patients). Pulmonary involvement occurs in 5–10% of pediatric cases. Hematologic complications such as hemolytic anemia, neutropenia, thrombocytopenia, aplastic anemia, agranulocytosis, and hemophagocytic syndrome can develop [14]. Neurological disorders, including encephalitis, meningitis, acute disseminated encephalomyelitis, acute cerebellar ataxia, facial nerve palsy, Alice in Wonderland syndrome and Guillain–Barre syndrome, occur in 1–5% of patients [15,16,17]. Myocarditis and pericarditis can also occur [13]. A study in Denmark found that EBV-associated IM is linked to a higher risk of malignancy, particularly hematologic malignancy, in the years following disease onset [18].

The mainstay of treatment for individuals with IM is supportive care [10]. The use of corticosteroids in the treatment of EBV-induced IM has been controversial [19]. However, a course of corticosteroids is warranted in individuals with impending airway obstruction [10].

Clinical manifestations and laboratory findings in pediatric patients (<18 years of age) with EBV-associated IM have been described in several retrospective cohorts from various geographic regions including Denmark (n = 95), Turkey (n = 66), Mexico (n = 163), Korea (n = 81), Northern China (n = 418) and China (n = 499) [20,21,22,23,24,25]. The spectrum of clinical signs and symptoms as well as the frequency and types of complications shown in these studies suggest a high degree of diversity in the clinical course of EBV-associated IM. Nevertheless, it should be emphasized that the studies are quite variable particularly regarding the age distribution of enrolled patients, which reflects differences in the epidemiology of EBV infection in particular areas. In addition, these studies reveal very different approaches to treatment in different parts of the world, particularly in the context of antiviral therapy. Importantly, no predictors of clinical complications associated with EBV-associated IM have been identified so far, in part due to the relatively small sample size in some of the clinical cohorts.

The aim of the present study was to analyze clinical characteristics and selected laboratory findings in a cohort of EBV-associated IM pediatric patients hospitalized in a national tertiary clinical center that specializes in pediatric infectious disease care in Croatia during the period of six consecutive years and to investigate the possible predictors of IM complications. To the best of our knowledge, this is the largest clinical study on EBV-associated IM in pediatric patients from Europe reported so far.

## 2. Materials and Methods

### 2.1. Study Design, Patients and Data Collection

Medical records of all patients aged 0–18 years who were hospitalized at the pediatric infectious diseases departments of the University Hospital for Infectious Diseases “Dr. Fran Mihaljević” (UHID) from January 2013 to December 2018 with a diagnosis of IM were collected. Reasons for admission were prolonged fever, confirmed or suspected bacterial co-infection, subjective airway obstruction, inability to maintain adequate enteral hydration/nutrition and social reasons. Patients with incomplete medical documentation were excluded from the study. The study included only patients with EBV-associated IM that was confirmed with serology (positive VCA IgM antibodies) and/or viremia. Clinical diagnosis in IM patients was based on the following signs and symptoms: fever, tonsillitis, lymphadenopathy and hepatosplenomegaly. We also analyzed laboratory parameters including white blood count, presence of atypical lymphocytes, levels of aminotransferases, bilirubin and lactate dehydrogenase (LDH), hemoglobin, platelet count, etc. We also analyzed the presence of a rash with or without the use of antibiotics, treatment strategy including the use of corticosteroids and duration of hospitalization. The Ethics committee of the University Hospital approved the study for Infectious Diseases, Zagreb, Croatia, on 28 August 2019 (no. 01-1247-3-2019).

### 2.2. Statistical Analysis

Data analysis and visualization were performed using R (version 4.1.1., R Core Development Team, Vienna, Austria) and ggplot2 package (Version 3.5.0). Qualitative variables were compared using the chi-square test and Fisher’s exact test where applicable. Pairwise comparisons were performed with the independent *t*-test or Mann–Whitney U test in the case of non-parametric distributions. Quantitative variables with more than two levels were compared with one-way ANOVA or Kruskal–Wallis test in case of non-parametric distributions. Correlation between quantitative variables was analyzed using Spearman’s correlation coefficient and the correlation test. All statistical tests were two-tailed with the significance level set at 95%. *p*-values were corrected for multiple testing with the Bonferroni method.

Multiple linear regression was used to analyze the independent correlation of hospitalization length with demographic, clinical and laboratory parameters. Predictor selection was performed with the best subset selection algorithm. Residual distribution normality was assessed using residual versus fit plots and quantile–quantile plots.

## 3. Results

### 3.1. Demographic and Clinical Data

During the 6-year study period, a total of 212 patients were hospitalized in UHID due to acute IM. The main characteristics of the study cohort are shown in Table 1. The median age of patients was 14.7 years (IQR 7.7–16.5 years) and 55.2% of patients were female. When considering age groups, 21.7% of patients were in the 0–5 years group, 18.9% of patients were in the 6–12 years group and 59.4% of patients were in the 13–18 years group. Girls were older (median age of 15.8 years) than boys (median age of 9.9 years). Acute EBV-associated IM was diagnosed based on positive VCA IgM findings in 203 (95.8%) patients (other serological markers included VCA IgM antibodies as well as EA-D and EBNA IgG antibodies) and with a positive real-time PCR result (EBV DNA quantification by using LightMix EBV assay, TIB Molbiol, Berlin, Germany, on LightCycler instruments 2.0 and 480 II Instruments, Roche Diagnostics, Basel, Switzerland) in 9 (4.2%) patients. Additionally, 51.2% of patients were hospitalized in the first week of the disease (median 7 days, IQR 4–10 days) and the median hospitalization duration was 9 days (IQR 7–11 days). None of the patients analyzed in this study developed chronic EBV infection or suffered a lethal outcome.

### 3.2. Clinical and Laboratory Characteristics

Clinical and laboratory findings of patients with EBV-associated IM have been analyzed, particularly the association between symptoms and complications of IM and laboratory findings. The analysis of symptoms described in IM patients is shown in Table 2.

The most common symptoms are displayed in Table 2. It is noteworthy that symptom distribution, maximal fever and fever duration did not differ among different age groups (*p* > 0.05).

Patients who suffered from tonsillitis were more likely to develop lymphadenopathy (OR = 9.72, 95% CI 3.76–26.55, *p* < 0.001), rash (OR = 5.34, 95% CI 1.26–48.10, *p* = 0.011), and splenomegaly (OR = 2.94, 95% CI 1.17–7.32, *p* = 0.018). Furthermore, patients with periglandular edema were more likely to experience lymphadenopathy (OR = 6.79, 95% CI 2.02–35.75, *p* < 0.001) and periorbital edema (OR = 4.58, 95% CI 2.18–9.78, *p* < 0.001). Patients who had hepatomegaly were more likely to present with splenomegaly (OR = 14.50, 95% CI 6.48–34.22, *p* < 0.001).

Patients who had hepatosplenomegaly exhibited a number of significantly higher laboratory findings than other patients, including maximal leukocyte count (medians 16.3 and 14.5 × 10^9^/L, *p* = 0.028), maximal ALT levels (medians 175.0 and 130.0 U/L, *p* = 0.017) and maximal LDH levels (medians 429.0 and 355.0 U/L, *p* = 0.006). We also found a weak positive correlation between hospitalization length and maximal AST levels (r = 0.30, *p* < 0.001) and LDH levels (r = 0.28, *p* = 0.002).

Analysis of complications described in IM patients is presented in Table 3. The most common complications included tonsillar hypertrophy (25.0%), thrombocytopenia (23.1%), anemia (18.4%), neutropenia (14.6%) and leukopenia (7.1%). Patients who had three or more complications had longer hospitalization lengths than other patients (medians 11 and 9 days, *p* = 0.049).

Patients who already developed anemia at admission or during the first week of hospitalization were more likely to exhibit splenomegaly (OR = 2.91, 95% CI 1.55–5.07, *p* = 0.002) and neutropenia (OR = 3.77, 95% CI 1.48–9.41, *p* = 0.004). Similarly, patients who had thrombocytopenia were more likely to present with neutropenia (OR = 2.58, 95% CI 1.04–6.27, *p* = 0.033).

We also analyzed a possible association between age and clinical presentation. Patients in the 0–6 years group were more likely to develop anemia than patients in the 13–18 years group (OR = 2.79, 95% CI 1.14–6.77, *p* = 0.01). On the contrary, patients in the 13–18 years group were more likely to develop thrombocytopenia than patients in the 7–12 years group (OR = 4.9, 95% CI 1.3–8.5, *p* = 0.001) and patients in the 0–5 years group (OR = 5.5, 95% CI 1.4–8.7, *p* < 0.001).

We also analyzed the laboratory findings in subsets of patients with anemia, thrombocytopenia, or neutropenia (Figure 1). Patients who presented with anemia had significantly higher levels of LDH (medians of 560.0 and 410.0 U/L, *p* = 0.004), higher ESR (medians of 43.5 and 29.0 mm/h, *p* = 0.007) and lower ANC (medians of 2.2 and 2.9 × 10^9^/L, *p* = 0.009) than other patients.

Furthermore, patients who had thrombocytopenia showed significantly higher levels of bilirubin (medians of 15.0 and 12.0 µmol/L, *p* = 0.005), LDH (medians of 572.0 and 405.0 U/L, *p* < 0.001) and lower ANC (medians of 2.19 and 2.88 × 10^9^/L, *p* < 0.001) than other patients. Patients who presented with thrombocytopenia showed longer hospitalization durations than other patients (medians of 10 and 9 days, *p* = 0.028).

Likewise, patients who exhibited neutropenia displayed lower levels of bilirubin (medians of 12.0 and 15.0 µmol/L, *p* = 0.040), higher levels of LDH (medians 558.0 and 412.0 U/L, *p* < 0.001) and lower ESR (medians 20.0 and 30.0 mm/h, *p* = 0.030).

In an attempt to investigate independent predictors of hospitalization length in our IM cohort, we applied multiple linear regression (Table 4). The analysis utilized demographic, clinical and laboratory parameters in predicting hospitalization length. The best subset selection method identified four parameters that most accurately predicted hospitalization length: platelet count, bilirubin, ESR and AST (Table 4). For each unit decrease in platelet count, hospitalization length increases by 0.12 days (95% CI 0.02–0.22, *p* = 0.039). For each unit increase in billirubin, hospitalization length increases by 0.08 days (95% CI 0.01–0.15, *p* = 0.044). For each unit increase in ESR, hospitalization length increases by 0.10 days (95% CI 0.01–0.19, *p* = 0.048). For each unit increase in AST, hospitalization length increases by 0.07 days (95% CI 0.01–0.14, *p* = 0.050). The model explains 30.2% of the variation in hospitalization length (adjusted R^2^ = 0.302).

The main therapeutic approaches were supportive care, corticosteroids and antibiotics. All patients received supportive care, while corticosteroids were administered to 124 (58.5%) patients (median disease day of admission = 9, IQR 6–12). The main indication for corticosteroid treatment was tonsillar hypertrophy (OR = 6.69, 95% CI 2.78–18.63, *p* < 0.001). Notably, patients who experienced lymphadenopathy (OR = 2.22, 95% CI 1.11–4.61, *p* = 0.024) and periglandular oedema (OR = 2.96, 95% CI 1.45–6.34, *p* = 0.002) were more likely to require corticosteroid treatment.

When considering antibiotics, 124 (58.5%) patients received antibiotic treatment. The majority of those patients were treated with third-generation cephalosporins (71.0%), coamoxiclav (12.1%), azithromycin (9.8%), phenoxymethylpenicillin (7.1%) or amoxicillin (3.8%). Patients treated with co-amoxiclav or amoxicillin were more likely to develop rash than other patients (OR = 5.24, 95% CI 2.18–13.02, *p* < 0.001). Interestingly, these patients were more likely to develop maculopapular rash than other patients who developed rash (OR = 3.81, 95% CI 1.12–16.34, *p* = 0.040).

## 4. Discussion

Age-specific analysis of EBV seroprevalence in the general population from Croatia ranges between 59.6% in children < 6 years of age and 98.3% in persons aged 30–39 years [26]. In patients with EBV-associated IM analyzed in this study, the peak incidence occurred at an age group of 13–18 years with a median age of 14.7 years. This finding is expected because in developed countries, infectious mononucleosis occurs more often in older children and adolescents [27]. However, some studies, including a Danish cohort of 95 children with EBV-associated IM, have shown an unexpected age distribution with a median age of 7 years despite the assumption that IM in younger children is mostly subclinical [20].

Patients with infectious mononucleosis most often present with fever, tonsillitis and cervical lymph node enlargement, which were also the most common clinical features presented in our study [27]. Fever was observed in almost all patients in our cohort with a median duration of 8 days, which is similar to the data observed in a Korean cohort described by Son et al. [23]. Contrary to the Korean cohort, our study failed to show any differences in the duration of fever among different age groups [23]. Proportions of febrile patients in cohorts of patients with EBV-associated IM reported in the literature are variable, with Balfour et al. (2005) and Rea et al. reporting only 30% and 45% febrile patients, respectively, while other studies reported > 80% of febrile patients [21,22,24,28,29,30]. In addition, there is a high degree of variability regarding the prevalence of tonsillitis ranging between 100% observed by Balfour et al. (2005), which is similar to our observations, and only 62.3% of patients reported by Topp et al. (2015) [20,28]. In a study by Topp et al. (2015) and Son et al. (2011), splenomegaly and hepatomegaly were much less frequent (16.8% and 23.6%; 12.3% and 24.7%, respectively) than in our study (73.1% and 77.4%), which could be due to clinical bias or the fact that we performed abdominal ultrasound much more frequently [20,23].

Some studies reported differences in the clinical presentation of IM according to age. Topp et al. (2015) divided the cohort into three age groups (0–4 years, 5–10 years and 11–15 years) and showed that young children typically presented with a runny nose, fever, fatigue and cervical adenitis (89.5%), while older age groups more often suffered from headache, sore throat, abdominal pain, nausea and myalgia/arthralgia [20]. Based on our study, Croatian children do not show differences in clinical findings based on age stratification. The logical explanation for our result is that EBV infection in younger age groups, especially infants, is often asymptomatic or subacute without a typical clinical presentation and therefore does not need hospital intervention, unlike patients enrolled in our study.

Regarding laboratory parameters, the majority of patients presented with leukocytosis, which is in accordance with observations from other clinical cohorts [20,21,22,23,24,25]. We also detected thrombocytopenia, anemia, neutropenia and leucopenia, which are well-established hematologic abnormalities in IM [31]. We noticed that younger children had lower levels of hemoglobin, and older children had lower platelet numbers. Patients with a more severe clinical course, e.g., with three or more IM complications, as well as patients presenting with thrombocytopenia had significantly longer hospitalization durations. Evaluation of other potential predictors of hospitalization duration including age distribution, symptoms and laboratory findings did not reveal significant results, except a weak positive correlation between hospitalization duration and maximal levels of AST and LDH.

Multiple linear regression analysis of demographic, clinical and laboratory parameters identified platelet count, bilirubin, ESR and AST as the most accurate predictors of hospitalization duration but this model had relatively low predictive power.

The incidence of airway obstruction in IM described in the literature ranges between 1 and 3.5% of patients [32,33]. Although none of the patients in our cohort experienced airway obstruction, due to possible complications associated with airway obstruction due to hypertrophic tonsils, a proportion of patients in our study were treated with corticosteroids.

Neurologic complications were present in three children, which is compatible with incidences found in the literature [10].

We also recorded individual cases of nephritis, myocarditis, pericarditis and hemophagocytic lymphohistiocytosis. EBV-induced pneumonitis is a very rare entity and is hardly seen among immunocompetent individuals [34].

The standard treatment approach to treatment in EBV-associated IM is symptomatic [10]. In a study by Thompson et al. (2005), corticosteroids were used in 44.7% of patients. Factors associated with the observed increase in corticosteroid use included a history of repeated visits, inpatient admission, and otorhinolaryngology consultation [35]. De Paor et al. (2016) performed a meta-analysis of seven randomized clinical trials with a total of 333 immunocompetent patients with EBV-associated IM treated with acyclovir, valomaciclovir and valacyclovir [36]. Statistically significant improvements were observed in the treatment group for 2 of the 12 outcomes but they were of limited clinical significance, and the overall quality of evidence was estimated to be very low. In our study, all patients received symptomatic treatment. About 60% of patients were treated with corticosteroids, mostly patients who had hypertrophic tonsils when airway obstruction was feared. Children who had lymphadenopathy and periglandular edema were also more likely to receive corticosteroids. No adverse effects of corticosteroid therapy were noted nor was there any suspicion of malignancy that prevented the administration of corticosteroids. As there was no control group in our study, objective effectiveness of corticosteroid therapy could not be assessed, although it was our clinical subjective impression that it was beneficial to the resolution of tonsillar hypertrophy. The use of corticosteroids did not affect the duration of hospitalization. None of our patients received any antiviral treatment.

Although IM is a viral disease and antibiotic treatment is not recommended, antibiotics have been administered in about 60% of patients in our cohort. The main indication was coinfection with group A streptococcus (GAS) in 10.9% of children and diagnosis of nasopharyngitis in 14 children and sinusitis in 5 children. Knowing that the incidence of GAS carriage varies from 2 to 40%, the question remains if the tonsillitis reported in the studied patients was indeed a GAS coinfection or just GAS carriage with tonsillitis as one of the clinical features of IM [37]. Patients treated with co-amoxiclav or amoxicillin were more likely to develop a rash than other patients (OR = 5.24, 95% CI 2.18–13.02, *p* < 0.001). These patients were more likely to develop a maculopapular rash than other patients who developed a rash. Patients in a study by Çağlar et al. (2019) were also more likely to develop a rash after antibiotic consumption [21].

Despite reporting the largest cohort of hospitalized children with EBV-associated IM from Europe so far, it is important to point out that our study has limitations including the retrospective nature of the study, inclusion of only hospitalized patients and the inability to standardize the reasons for admission. As for the latter, we could not influence this, and the former was due to a significant lack of medical documentation (missing or inadequate medical histories, lacking or incomplete laboratory tests results, lack of information on treatment) in outpatient care of pediatric patients with IM. Therefore, it is our opinion that the inclusion of those patients would not give an objective and noteworthy observation. Taking into consideration the variability of data on demographic, clinical and laboratory parameters in children with EBV-associated described so far in several large international cohorts, a prospective study with a standardized diagnostic and clinical approach that includes a follow-up examination would be warranted to assess the burden of disease more objectively (particularly in younger age groups) and to characterize clinical/laboratory predictors of complications more extensively.

## Figures and Tables

**Figure 1 pathogens-14-00374-f001:**
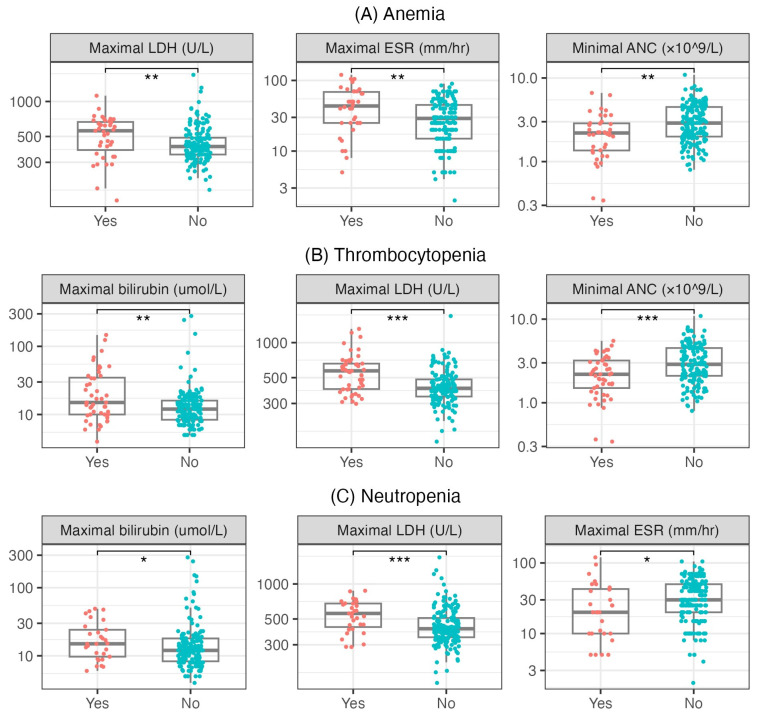
(**A**) Distributions of maximal levels of LDH, ESR and ANC regarding anemia. (**B**) Distributions of maximal levels of bilirubin, LDH and ANC regarding thrombocytopenia. (**C**) Distributions of maximal levels of bilirubin, LDH and ESR regarding neutropenia. The boxes show the median and interquartile range of the distribution, while the whiskers extend to the minimum and maximum non-outlier values of the distribution. Points denote individual patients. Mann–Whitney U test, *: *p* < 0.05, **: *p* < 0.01, ***: *p* < 0.001, *p*-values corrected with the Bonferroni method. LDH = lactate dehydrogenase, ESR = erythrocyte sedimentation rate, ANC = absolute neutrophil count, AST = aspartate transaminase.

**Table 1 pathogens-14-00374-t001:** Patients’ general characteristics. Numerical variables are expressed in median and interquartile range, while categorical variables are represented as absolute numbers and percentages of total patients.

Characteristic	
Age (years)	14.7 (7.7–16.5)
Male sex	95 (44.8%)
Age group (years)	
0–5	46 (21.7%)
6–12	40 (18.9%)
13–18	126 (59.4%)
Diagnosis	
VCA IgM	203 (95.8%)
RT-PCR	9 (4.2%)
Day of disease	7 (4–10)
Hospitalization length (days)	9 (7–11)

VCA = viral capsid antigen, RT-PCR = real-time polymerase chain reaction.

**Table 2 pathogens-14-00374-t002:** Number of symptoms. Numerical variables are expressed in median and interquartile range, while categorical variables are represented as absolute numbers and percentages of total patients.

Symptom	N (%)
Tonsillitis	185 (87.3%)
Speckled exudate	121 (65.1%)
Whitewash exudate	34 (16.0%)
Hyperemic tonsils	29 (13.7%)
Lymphadenopathy	168 (79.2%)
Bilateral cervical	115 (54.3%)
Unilateral cervical	44 (20.8%)
Multiple regions	9 (4.3%)
Hepatomegaly	164 (77.4%)
Splenomegaly	155 (73.1%)
Hepatosplenomegaly	141 (66.6%)
Respiratory symptoms	125 (59.0%)
Periglandular edema	59 (27.8%)
Rash	58 (27.4%)
Maculopapular	27 (12.7%)
Macular	13 (6.1%)
Urticaria	8 (3.8%)
Other	10 (4.7%)
Periorbital edema	46 (21.7%)
Fever	206 (97.16%)
Maximal fever (°C)	39.2 (38.8–39.8)
Fever duration (days)	8 (5–10)

**Table 3 pathogens-14-00374-t003:** Number of complications. Percentages refer to the total number of patients.

Complications	N (%)
Tonsillar hypertrophy	53 (25.0%)
Thrombocytopenia	49 (23.1%)
Anemia	39 (18.4%)
Neutropenia	31 (14.6%)
Leucopenia	15 (7.1%)
Neurological complications	3 (1.4%)
Edema of the uvula	3 (1.4%)
Nephritis	1 (0.5%)
Myocarditis	1 (0.5%)
Pericarditis	1 (0.5%)
Hemophagocytic lymphohistiocytosis	1 (0.5%)

**Table 4 pathogens-14-00374-t004:** Multiple linear regression model predicting hospitalization length in the analyzed IM patients.

Predictor	Coefficient	Standard Error	*t*	*p*
Platelet count	−0.12	0.005	2.4	0.039
Billirubin	0.08	0.040	2.0	0.044
ESR	0.10	0.050	2.0	0.048
AST	0.07	0.040	1.8	0.050

ESR = erythrocyte sedimentation rate, AST = aspartate transaminase.

## Data Availability

Data available upon reasonable request from the corresponding author.

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
