# Peer review of "Clinical Features and Laboratory Findings of Hospitalized Children with Infectious Mononucleosis Caused by Epstein–Barr Virus from Croatia"

_pathogens, 2025, doi:10.3390/pathogens14040374_

Round 1

Reviewer 1 Report (Previous Reviewer 1)

Comments and Suggestions for Authors

The manuscript has been improved and solved my concerns. I don't have any comments for the current version of the manuscript.

Author Response

Reviwer 1.The manuscript has been improved and solved my concerns. I don't have any comments for the current version of the manuscript.

Authors response.Thank you very much for considering our work and helping us improve it.

Reviewer 2 Report (Previous Reviewer 2)

Comments and Suggestions for Authors

The authors provide an updated version of the manuscript which includes now a multiple linear regression model. The authors are able to identify 3 biological parameter that could predict the duration of hospitalisation: platelet count, bilirubin and erythrocyte sedimentation rate. The line 264 have to modified in that sense (AST is not predictive).

Author Response

Reviewer:

The authors provide an updated version of the manuscript which includes now a multiple linear regression model. The authors are able to identify 3 biological parameter that could predict the duration of hospitalisation: platelet count, bilirubin and erythrocyte sedimentation rate. The line 264 have to modified in that sense (AST is not predictive).

Author's response: In order to present the data on correlation analysis results and multiple regression, we modified/added text in section results (176-178) and discussion (298-304). We hope this clarify the issue of hospitalization duration predictors.

Reviewer 3 Report (New Reviewer)

Comments and Suggestions for Authors

The article entitled “Clinical and Laboratory Characteristics of Pediatric Infectious Mononucleosis: A Single-Center Study” analyzes the clinical and laboratory features of pediatric infectious mononucleosis (IM) in a Croatian cohort, providing valuable epidemiological data on the largest hospitalized pediatric population with EBV-associated IM in Europe. While the study presents a thorough dataset, its novelty beyond sample size is limited, and additional statistical analysis could strengthen the conclusions.

Major:

  1. Novelty & Significance
  • The study provides important epidemiological data, but does not present fundamentally new findings beyond confirming trends observed in previous studies.
  • Suggestion: The authors could highlight any unique clinical patterns in Croatian children compared to other European cohorts. Additionally, a more in-depth discussion of how these findings impact clinical practice would be beneficial.
  1. Risk Factors & Predictors of Complications
  • The study identifies thrombocytopenia as associated with longer hospitalization, but does not establish strong predictive markers for complications.
  • Suggestion: A multivariate regression analysis incorporating age, sex, laboratory parameters, and clinical presentation could help identify independent predictors of severe disease or prolonged hospitalization.
  1. Clinical Implications & Treatment Considerations
  • The conclusion that "clinical assessment remains the most reliable parameter" is expected, but does not provide actionable recommendations.
  • Corticosteroid use is mentioned, but the study does not analyze its effectiveness or its relationship with clinical outcomes.
  • Suggestion: The authors could provide more detailed treatment data and discuss whether corticosteroid use influenced disease severity or duration of hospitalization.

Minor:

  1. Discussion & Literature Comparison
  • A more detailed comparison with similar European pediatric cohorts would add depth.
  • Suggestion: Discuss any unique epidemiological, clinical, or laboratory findings in the Croatian population compared to other regions.
  1. Statistical Analysis & Data Presentation
  • Consider additional statistical validation for severe vs. mild cases.
  • Suggestion: If available, the authors could stratify data by age groups or use logistic regression models to determine factors associated with severe disease.
Comments on the Quality of English Language
  • Minor language refinements needed for clarity.

         Line 217; 'analysed' must be 'analyzed'.

         Line 271; 'lymphohysticytosis' must be 'lymphohistiocytosis'.

  • Suggestion: A professional language edit or proofreading could enhance readability, particularly in the discussion and conclusion sections.

Author Response

Reviewer's comments

  1. Novelty & Significance
  • The study provides important epidemiological data, but does not present fundamentally new findings beyond confirming trends observed in previous studies.
  • Suggestion:The authors could highlight any unique clinical patterns in Croatian children compared to other European cohorts. Additionally, a more in-depth discussion of how these findings impact clinical practice would be beneficial.

Author's response:

In section discussion, we added a text discussing the differences in clinical patterns of our cohort versus Topp et al (Danish cohort, reference 20, lines 283-292). Modified text was highlighted in red.  

  1. Risk Factors & Predictors of Complications
  • The study identifies thrombocytopenia as associated with longer hospitalization, but does not establish strong predictive markers for complications.
  • Suggestion: A multivariate regression analysis incorporating age, sex, laboratory parameters, and clinical presentation could help identify independent predictors of severe disease or prolonged hospitalization.

Author's response: Multiple linear regression model results have allready been included in the original version (Table 4). We clarified the analysed parameters in the modified text (224-240) and added a reference to these results in section abstract (27-29) and discussion (301-303).  

Modified text in section Results:

In an attempt to investigate independent predictors of hospitalization length in our IM cohort, we applied multiple linear regression (Table 4). The analysis utilized demographic, clinical and laboratory parameters in predicting hospitalization length. The best subset selection method identified four parameters that most accurately predicted hospitalization length: platelet count, bilirubin, ESR and AST (Table 4). For each unit decrease in platelet count, hospitalization length increases by 0.12 days (95%CI 0.02-0.22, p = 0.039). For each unit increase in billirubin, hospitalization length increases by 0.08 days (95%CI 0.01-0.15, p = 0.044). For each unit increase in ESR, hospitalization length increases by 0.10 days (95%CI 0.01-0.19, p = 0.048). For each unit increase in AST, hospitalization length increases by 0.07 days (95%CI 0.01-0.14, p = 0.050). The model explains 30.2% of the variation in hospitalization length (adjusted R2 = 0.302).

Table 4. Multiple linear regression model predicting hospitalization length in the                          analyzed IM patients.

Predictor

Coefficient

Standard Error

t

p

Platelet count

-0.12

0.005

2.4

0.039

Billirubin

0.08

0.040

2.0

0.044

ESR

0.10

0.050

2.0

0.048

AST

0.07

0.040

1.8

0.050

                                ESR = erythrocyte sedimentation rate, AST = aspartate transaminase

  1. Clinical Implications & Treatment Considerations
  • The conclusion that "clinical assessment remains the most reliable parameter" is expected, but does not provide actionable recommendations.
  • Corticosteroid use is mentioned, but the study does not analyze its effectiveness or its relationship with clinical outcomes.
  • Suggestion:The authors could provide more detailed treatment data and discuss whether corticosteroid use influenced disease severity or duration of hospitalization.

Author's response:

In section Results, we added a text describing the treatment data: The main therapeutic approaches were supportive care, corticosteroids and antibiotics. All patients received supportive care, while corticosteroids were administered to 124 (58.5%) patients (median disease day of admission = 9, IQR 6-12). The main indication for corticosteroid treatment was tonsillar hypertrophy (OR = 6.69, 95% CI 2.78-18.63, p<0.001). Notably, patients who experienced lymphadenopathy (OR = 2.22, 95% CI 1.11-4.61, p = 0.024) and periglandular oedema (OR = 2.96, 95% CI 1.45-6.34, p = 0.002) were more likely to require corticosteroid treatment. When considering antibiotics, 124 (58.5%) patients received antibiotic treatment. The majority of those patients were treated with third-generation cephalosporins (71.0%), co-amoxiclav (12.1%), azithromycin (9.8%), phenoxymethylpenicillin (7.1%) or amoxicillin (3.8%). Patients treated with co-amoxiclav or amoxicillin were more likely to develop rash than other patients (OR = 5.24, 95% CI 2.18-13.02, p<0.001). Interestingly, these patients were more likely to develop maculopapular rash than other patients who developed rash (OR = 3.81, 95% CI 1.12-16.34, p = 0.040).

Minor:

  1. Discussion & Literature Comparison
  • A more detailed comparison with similar European pediatric cohorts would add depth.
  • Suggestion:Discuss any unique epidemiological, clinical, or laboratory findings in the Croatian population compared to other regions.

Author's response:

In section discussion, we added a text discussing the differences in clinical patterns of the Danish cohort (European study by Topp et al).    

  1. Statistical Analysis & Data Presentation
  • Consider additional statistical validation for severe vs. mild cases.
  • Suggestion:If available, the authors could stratify data by age groups or use logistic regression models to determine factors associated with severe disease.

Author's response:

In section Results, the following text reffers to the issue of age: We also analyzed a possible association between age and clinical presentation. Pa-tients in the 0-6 years group were more likely to develop anemia than patients in the 13-18 years group (OR = 2.79, 95% CI 1.14-6.77, p = 0.01). On the contrary, patients in the 13-18 years group were more likely to develop thrombocytopenia than patients in the 7-12 years group (OR = 4.9, 95% CI 1.3-8.5, p = 0.001) and patients in the 0-5 years group (OR = 5.5, 95% CI 1.4-8.7, p < 0.001).

In section Results, the issue of disease severity is analysed by analysing the type and the frequency of complictions (Table 3) and the text: Analysis of complications described in IM patients is presented in Table 3. The most common complications included tonsillar hypertrophy (25.0%), thrombocytopenia (23.1%), anemia (18.4%), neutropenia (14.6%) and leukopenia (7.1%). Patients who had three or more complications had longer hospitalization lengths than other patients (me-dians 11 and 9 days, p = 0.049).

In section discussion, we reffer to the the issue in the text: Patients with more severe clinical course, e.g. with three or more IM complications as well as patients presenting with thrombocytopenia, had significantly longer hospitalization duration.

Comments on the Quality of English Language

  • Minor language refinements needed for clarity.

         Line 217; 'analysed' must be 'analyzed'.- corrected

         Line 271; 'lymphohysticytosis' must be 'lymphohistiocytosis'- corrected.

  • Suggestion:A professional language edit or proofreading could enhance readability, particularly in the discussion and conclusion sections.- performed

This manuscript is a resubmission of an earlier submission. The following is a list of the peer review reports and author responses from that submission.

Round 1

Reviewer 1 Report

Comments and Suggestions for Authors

This study is to determine the correlation between demographic, laboratory and clinical features among young patients with EBV-associated infectious mononucleosis and identify potential predictors of complications. The analysis results demonstrate clinical assessment is the most reliable parameter and the role of laboratory data is limited. This manuscript is well drafted with clear presentation, there are only few tiny issues to be solved.

1.        In line 55, there is no reference number in the brackets.

2.        In line 71, … [for review see 13] … can be replaced with … [13] …

3.        In line 96, …(n=418) …, a bracket is missing. In addition, an extra full stop dot should be deleted.

4.        In line 130, what does “(22)” represent? Should it be deleted?

5.        A straight line is missing before line 160.

6.        In line 243, …week… should be corrected as …weak…

7.     In line 326, a full stop is missing.

Author Response

Reviewer 1.

Dear Reviewer 1., thank you very much for considering our work and points that you have raised.

  1. In line 55, there is no reference number in the brackets.
  2. In line 71, … [for review see 13] … can be replaced with … [13] …
  3. In line 96, …(n=418) …, a bracket is missing. In addition, an extra full stop dot should be deleted.
  4. In line 130, what does “(22)” represent? Should it be deleted?
  5. A straight line is missing before line 160.
  6. In line 243, …week… should be corrected as …weak…
  7. In line 326, a full stop is missing.

All issues have been reviewed by co-authors and corrected as instructed (changes are indicated in red).

Reviewer 2 Report

Comments and Suggestions for Authors

Prtoric et al. propose a clinical and biological description of n=212 patients with infectious mononucleosis in paediatric hospitalised patients. This retrospective study is well documented, and all clinical symptoms are reported, together with routine biological results (leukocyte counts, formula, hemoglobin, platelets, ALT, AST, LDH, bilirubin). This is a complete description of severe IM in children. However, several points need to be stressed and prevent the publication in this form.

Major points:

-          The main point of this study is to predict the severity of the symptoms/biological findings. However, the initial symptoms /biological findings at the time of hospitalisation are not correlated to the 1 or 2-Week severity. It should have been useful for the clinician to identify the initial finding that could predict further severe severity and in fine could guide the hospital discharge of the patient. The authors conclude that patients who developed three or more complications and those with thrombopenia showed longer hospitalisation, but which are the complications and what are the cause for longer hospitalisation? A statistical model could have been used to identify the predictive factors for IM complications.

-          Line 49: the sentence with MS is off topic.

-          Line 202: the laboratory findings reported here are descriptive but not predictive of severity of the infectious mononucleosis.

Minor points:

-          The introduction could be summarized (line 67-87).

-          The sentence line 141 needs to be remove.

-          Line 156: what are the causes for hospitalisation?

-          The text line 167 to 177 is redundant with table 2.

-          Line 192: ‘patients who developed anemia’: at which delay from hospitalisation?

-          The discussion is far too long.

Author Response

Reviewer 2

Major points:

-          The main point of this study is to predict the severity of the symptoms/biological findings. However, the initial symptoms /biological findings at the time of hospitalisation are not correlated to the 1 or 2-Week severity. It should have been useful for the clinician to identify the initial finding that could predict further severe severity and in fine could guide the hospital discharge of the patient. The authors conclude that patients who developed three or more complications and those with thrombopenia showed longer hospitalisation, but which are the complications and what are the cause for longer hospitalisation? A statistical model could have been used to identify the predictive factors for IM complications.

-          Line 49: the sentence with MS is off topic

-          Line 202: the laboratory findings reported here are descriptive but not predictive of severity of the infectious mononucleosis.

 Minor points:

-          The introduction could be summarized (line 67-87).

-          The sentence line 141 needs to be remove.

-          Line 156: what are the causes for hospitalisation?

-          The text line 167 to 177 is redundant with table 2.

-          Line 192: ‘patients who developed anemia’: at which delay from hospitalisation?

-          The discussion is far too long.

Author’s response:

Dear Reviewer 2!

Thank you for the valued review and chance to enhance our study. You input is appreciated and we strive to enhance the quality our study!

The aim of the study was to analyse clinical and laboratory characteristics of hospitalized children with IM and investigate possible predictors of IM complications. Complications noted in our study are described in Table 3. Our study states that most common complication was tonsillar hypertrophy as well as a notion that patients that presented with thrombocytopenia or had at least three of the listed complications (regardless of which ones) were predictors of longer hospitalization. Besides thrombocytopenia, no other laboratory parameter or complication was a singular predictor for longer hospitalization.  Although our study tries to predict the risk for complications and longer hospital stay, unfortunately does not give an algorithm on management of IM.

All other comments are regarded in the manuscript and the added text is marked in red. In addition, parts of the text have been reduced in introduction and discussion (this was balanced with respect to comments from other reviewers). 

Reviewer 3 Report

Comments and Suggestions for Authors

Overall comments

This manuscript provides an overview of Epstein-Barr Virus infection and infectious mononucleosis, and describes the clinical and laboratory findings in a cohort of children admitted to a single hospital in Zagreb, Croatia. The manuscript was submitted for consideration for inclusion in a Special Issue of Pathogens, “Emerging and Neglected Pathogens in the Balkans”. The study reports data from collected over five years, from 2013-2018.

The paper provides reasonable data from a relatively large cohort of patients in Croatia.

Major

The review is reasonably well done. However, there needs to be some clarification. This study only concerns patients who were admitted to the hospital. What criteria were used to determine admission? In many places, admission for EBV would be unusual, as this is usually managed on an outpatient basis.

What fraction of patients presenting with EBV were hospitalized and included in this study? Do the clinical characteristics of the hospitalized patients differ from the non-hospitalized patients? I would be helpful to include the data from both non-hospitalized and non-hospitalized groups, since this would give a much more complete picture of the disease.

What treatments were the patients described in the paper given? The paper mentions treatments that were used in other cohorts and the benefits and risks of those treatments, but does not describe treatments given to the patients described. Does the experience with this cohort support or weigh against certain treatments, like steroids, discussed for other cohorts, which can be problematic if a patients doesn’t actually have acute EBV infection, but instead has something else, like leukemia/lymphoma.

Minor

Typos:

line 134, Kruskal-Walli’s

Introduction

The Introduction provides a reasonable overview of EBV and infectious mononucleosis in children.

Methods

The paper needs to provide more detail about the methods, particularly the methods used to diagnose EBV infection, since that is the point of this paper. This is true both for the immunological and viral load studies.

Results

The results are reasonable and consistent with other EBV cohorts.

Discussion

The discussion is reasonable, placing the description of this cohort in the context of other cohorts and knowledge about the disease in general.

Author Response

Major

The review is reasonably well done. However, there needs to be some clarification. This study only concerns patients who were admitted to the hospital. What criteria were used to determine admission? In many places, admission for EBV would be unusual, as this is usually managed on an outpatient basis.

 What fraction of patients presenting with EBV were hospitalized and included in this study? Do the clinical characteristics of the hospitalized patients differ from the non-hospitalized patients? I would be helpful to include the data from both non-hospitalized and non-hospitalized groups, since this would give a much more complete picture of the disease.

 What treatments were the patients described in the paper given? The paper mentions treatments that were used in other cohorts and the benefits and risks of those treatments, but does not describe treatments given to the patients described. Does the experience with this cohort support or weigh against certain treatments, like steroids, discussed for other cohorts, which can be problematic if a patients doesn’t actually have acute EBV infection, but instead has something else, like leukemia/lymphoma.

 Minor

Typos:

line 134, Kruskal-Walli’s

Methods

The paper needs to provide more detail about the methods, particularly the methods used to diagnose EBV infection, since that is the point of this paper. This is true both for the immunological and viral load studies.

Dear Reviewer 3!

Thank you for your insightful review! Our study only concerns patients who were hospitalized. The reasons for admission were prolonged fever, confirmed or suspected bacterial co-infection, subjective airway obstruction, inability to maintain adequate enteral hydration/nutrition and social reasons (added to the manuscript). We were unable to objectivize further as this was unfortunately a retrospective study and we could not have dictated the reasons for admission. Also, as we added to the manuscript as a limitation of our study, we are unable to compare characteristics (both clinical and laboratory) due to general lacking medical documentation of non-hospitalized children (mostly lack of clinical assessment of symptoms at presentation by physician and fewer laboratory tests results as well as no follow-up; lacking information on treatment).

The treatment given to our patients has been added to the manuscript – all the patients were given symptomatic treatment, none received any antiviral therapy, 60% received corticosteroid therapy and 60% systematic antibiotic treatment during the course of disease. We have not observed any adverse reactions due to corticosteroid therapy. None of our patients had been diagnosed or been presented as a suspicion of leukemia / lymphoma / other malignancies during the hospitalization.

Additional comment: The paper needs to provide more detail about the methods, particularly the methods used to diagnose EBV infection, since that is the point of this paper. This is true both for the immunological and viral load studies.

Authors response: In section Materials and methods we included a more detailed description of serological and molecular methods used for the etiological diagnostics of EBV infection as a standerd-of-care approach in our institution.

All the changes in the text have been highlighted in red.

Round 2

Reviewer 2 Report

Comments and Suggestions for Authors

The main results are not explained sufficiently to the reader. A figure could be informative.

  • A statistical model could have been used to identify the predictive factors for IM complications.

The discussion is far too long. 

Author Response

Responses to reviewer 2.

Main points from reviewer 2.

  1. The main results are not explained sufficiently to the reader. A figure could be informative.
  2. A statistical model could have been used to identify the predictive factors for IM complications.
  3. The discussion is far too long. 

Author's response:

In an attempt to investigate predictors of hospitalization lenght in the IM cohort, we constructed a multiple linear regression model as suggested by the reviewer. In sections Material and methods we included additional text on the methodology, in section results we included Table 4 containing results of multiple linear regression and a text explaining the results to the reader and we also added a comment in discussion. As recommended, we significantly reduced discussion while trying to keep the main points of the text.

Reviewer 3 Report

Comments and Suggestions for Authors

I believe that the authors made adequate responses to the comments in the review.

Author Response

Responses to reviewer 3

Reviewer's comment:

I believe that the authors made adequate responses to the comments in the review.

Authors response: Thank you very much for your time and effort invested in this review.
